# Resolving temperature limitation on spring productivity in an evergreen conifer forest using a model-data fusion framework

Stephanie G. Stettz[1], Nicholas C. Parazoo[2], A. Anthony Bloom[2], Peter D. Blanken[3], David R. Bowling[4], Sean P. Burns[3,5], Cédric Bacour[6], Fabienne Maignan[7], Brett Raczka[5] , Alexander J. Norton[2], Ian Baker[8], Mathew Williams[9,10], Mingjie Shi[11], Yongguang Zhang[12], Bo Qiu[12]

[1] Department of Earth System Science, University of California Irvine, Irvine, California, USA
[2] Jet Propulsion Laboratory, California Institute of Technology, Pasadena, California, USA
[3] Department of Geography, University of Colorado Boulder, Boulder, Colorado, USA
[4] School of Biological Sciences, University of Utah, Salt Lake City, Utah, USA
[5] National Center for Atmospheric Research, Boulder, Colorado, USA
[6] NOVELTIS, 153 rue du Lac, 31670 Labège, France
[7] Laboratoire des Sciences du Climat et de l'Environnement, LSCE/IPSL, CEA-CNRS-UVSQ, Université Paris-Saclay, Gif-sur-Yvette, France
[8] Cooperative Institute for Research in the Atmosphere, Colorado State University, Fort Collins, Colorado, USA
[9] School of GeoSciences and National Centre for Earth Observation, University of Edinburgh, Edinburgh, UK
[10] National Centre for Earth Observation, Edinburgh EH9 3FF, Edinburgh, UK
[11] Pacific Northwest National Laboratory, 902 Battelle Blvd, Richland, WA 99354
[12] International Institute for Earth System Sciences, Nanjing University, Nanjing, Jiangsu Province, China

*Correspondence to:* Stephanie Stettz (sstettz@uci.edu)

**Abstract**

The flow of carbon through terrestrial ecosystems and the response to climate is a critical but highly uncertain process in the global carbon cycle.  However, with a rapidly expanding array of in situ and satellite data, there is an opportunity to improve our mechanistic understanding of the carbon (C) cycle's response to land use and climate change. Uncertainty in temperature limitation on productivity poses a significant challenge to predicting the response of ecosystem carbon fluxes to a changing climate.  Here we diagnose and quantitatively resolve environmental limitations on growing season onset of gross primary production (GPP) using nearly two decades of meteorological and C flux data (2000-2018) at a subalpine evergreen forest in Colorado, USA. We implement the CARDAMOM model-data fusion network to resolve the temperature sensitivity of spring GPP. To capture a GPP temperature limitation—a critical component of integrated sensitivity of GPP to temperature—we introduced a cold temperature scaling function in CARDAMOM to regulate photosynthetic productivity.  We found that GPP was gradually inhibited at temperature below 6.0 °C ($\pm$ 2.6 °C) and completely inhibited below -7.1 °C ($\pm$ 1.1 °C).  The addition of this scaling factor improved the model's ability to replicate spring GPP at interannual and decadal time scales (r = 0.88), relative to the nominal CARDAMOM configuration (r = 0.47), and improved spring GPP model predictability outside of the data assimilation training period (r = 0.88) . While cold temperature limitation has an important influence on spring GPP, it does not have a significant impact on integrated growing season GPP, revealing that other environmental controls, such as precipitation, play a more important role in annual productivity.

This study highlights growing season onset temperature as a key limiting factor for spring growth in winter-dormant
evergreen forests, which is critical in understanding future responses to climate change.
**1. Introduction**
Northern hemisphere evergreen forests contribute significantly to terrestrial carbon (C) storage and exchange
(Beer et al., 2010; Thurner et al., 2014). High-latitude and high-elevation evergreen forests show increasing gross
primary productivity (GPP) with increasing temperature driven in large part by earlier growing seasons (Myneni et
al., 1997; Randerson et al., 1999; Forkel et al., 2016; Winchell et al., 2016, Lin et al., 2017). However, the response
of gross and net C fluxes to warming remains uncertain, especially in subalpine temperate forests, which can
experience freezing temperature while still absorbing large amounts of sunlight; both these factors ultimately
influence the timing and magnitude of GPP (Bowling et al., 2018). In particular, warmer springs can also lead to
earlier snowmelt, which can reduce spring C uptake through increased surface exposure to colder ablation-period air
temperatures (Winchell et al., 2016), and can reduce summer C uptake via drought (Hu et al., 2010). Many
subalpine forests in western North America are also highly water limited, with warming and earlier snow melt
creating accumulated water deficits, increased drought stress, and growing season C uptake losses (Wolf et al.,
2016; Sippel et al., 2017; Buermann et al., 2018, Goulden and Bales, 2019); these factors ultimately make subalpine
forest ecosystems sensitive to the direct and indirect effects of climate change and other disturbances, including the
effects of droughts, fires and insect infestations (Keenan et al., 2014; Frank et al., 2014; Knowles et al., 2015). The
uncertainty in the temperature sensitivity of springtime GPP, increasing vulnerability to disturbance, and GPP
modeling challenges (Anav et al., 2015) create urgency to improve our ability to observe and model these
ecosystems to understand how C exchange will be altered in a warming climate.
Fortunately, availability of long term ecosystem observations is improving. The expansion of international
flux tower networks over the last three decades (e.g. AmeriFlux, FLUXNET, ChinaFLUX, ICOS) has greatly
improved C flux sampling across global ecosystems at 1 km scale (Baldocchi 2008; Baldocchi et al., 2018), and the
number of spaceborne sensors continues to grow, allowing global estimation of gross primary production (GPP) and
net ecosystem C exchange (NEE) over the last decade (e.g. Stavros et al., 2017; Sun et al., 2017; Schimel et al.,
2019). While uncertainties in estimating C fluxes from in situ and satellite data remain a challenge, the expanding
observational record offers a great opportunity to study the temperature sensitivity of subalpine forests at multiple
temporal scales.
The range of modeling tools available to quantify and study major C pools under ever growing
observational constraints is also increasing. Process-based models, in general terms, use explicit mathematical
relationships to mechanistically describe bio-physical processes (Korzukhin et al., 2011; Huxman et al., 2003;
Keenan et al., 2012). In contrast, model-data fusion (MDF) is a relatively new tool that alters model parameters to
statistically reduce mismatches between observations and model predictions (Raupach et al., 2005; Wang et al.,
2009; Keenan et al., 2012). MDF methods can be used to statistically represent the terrestrial C balance by
generating optimized state and process variable parameterizations, with uncertainties, which best match the signal
and noise in observations (Bloom et al., 2020).

Models of varying complexity and assimilation capabilities have been used to study how C exchange varies with temperature in subalpine evergreen ecosystems (e.g., Moore et al., 2008; Scott-Denton et al., 2013; Knowles et al., 2018). Moore et al. (2008) used a simplified ecosystem function model and assimilated C flux data from the Niwot Ridge (US-NR1) subalpine evergreen forest AmeriFlux tower in Colorado to show the importance of accurate meteorological forcing for parameter optimization and the usefulness of assimilating C flux data for determining connections between the C and water cycles. Scott-Denton et al. (2013) integrated meteorological and flux data from 1999-2008 from the same site with an ensemble of more sophisticated Earth System Models (ESM) and showed higher rates of C uptake by the end of the 21$^{st}$ century associated with warming and lengthening growing seasons, and driven by greater increases of spring GPP relative to late season respiration.

Interestingly, model and empirical studies of the C flux response to climate at US-NR1 focus on the 2000-2011 period, which saw increasing summer drought coupled with sustained declines in spring temperature and GPP. US-NR1 has since experienced a gradual recovery of spring GPP with increased spring warming throughout 2011-2018 (Fig. 1), which begs the question: what is the temperature sensitivity of spring GPP over multiple decades of spring cooling and warming at US-NR1, and how well can data-constrained models reproduce long term variability? To answer this question, we combine a mechanistic ecosystem C model (Data Assimilation Linked Ecosystem Carbon, or DALEC2; Williams et al., 2005; Bloom et al., 2016) with the CARbon DAta-MOdel fraMework (CARDAMOM; Bloom and Williams, 2015; Bloom et al., 2020) driven by observed meteorological forcing and constrained against eddy covariance fluxes at US-NR1, to investigate the temperature sensitivity of this subalpine evergreen forest at seasonal and interannual timescales. We introduce a new cold temperature limitation function, trained on observed temperature, for more realistic simulation of spring GPP onset. The use of high quality and long term (2000-2018) meteorology and partitioned GPP data at US-NR1 to drive and constrain the model enables robust statistical analysis of interannual variability (IAV), and assessment of "model predictability" through training and validation against subsets of data. We also leverage a recent model intercomparison study (Parazoo et al., 2020), driven by site level meteorological data at US-NR1, to provide a model benchmark assessment, and extract any common environmental controls on modeled GPP. Finally, we examine whether using a decade of flux tower-derived GPP observations to train the model is sufficient to match and predict seasonal to annual patterns in GPP. Given the complexity of carbon-water cycle interactions during the growing (summer) season in this highly water limited ecosystem, and the relatively weak correlation between tower-derived spring and summer GPP ($r = -0.31$, $p = 0.20$), we focus on spring GPP-temperature interactions, with the aim to resolve just one piece of the larger, complex problem of understanding changes in C uptake in a subalpine evergreen ecosystem.

## 2. Materials & Methods

### 2.1. Study Site: Niwot Ridge, CO., USA

Our study focuses on an AmeriFlux (https://ameriflux.lbl.gov/) core site in Niwot Ridge, Colorado, USA (US-NR1, 40°1'58''N; 105°32'47'' W), where a tower-based eddy covariance system has been used to continuously monitor the net ecosystem exchange (NEE) of carbon dioxide over a subalpine forest since November 1998. The 26

m tall tower is located in a high elevation (3050 m) subalpine site in the Rocky Mountains of Colorado (Monson et al., 2002). Located in an evergreen needleleaf (ENF) ecosystem, the dominant tree species include lodgepole pine (*Pinus contorta*), subalpine fir (*Abies lasiocarpa*), and Engelmann spruce (*Picea engelmannii*) (Turnipseed et al., 2002; Turnipseed et al., 2004). Average annual precipitation is 800 mm, with a majority of precipitation falling in the winter as snow (Greenland, 1989; Knowles et al., 2015), which creates a persistent winter snowpack from November through early June (Bowling et al., 2018).

## 2.2. Observations

NEE measurements are screened for calm conditions using the standard $u_{star}$ filtering, gap-filled, and partitioned into GPP and ecosystem respiration based on the relationship between nighttime NEE (photosynthetically active radiation, PAR < 50 $\mu$mol m$^{-2}$ s$^{-1}$) and air temperature (Reichstein et al., 2005; Wutzler et al., 2018). Monthly averages of GPP based on nighttime partitioning show similar seasonal structure to results found using an alternative daytime partitioning algorithm (Lasslop et al., 2009), so only nighttime partitioned GPP data are reported here. All GPP estimates are processed as half hourly means, then averaged monthly. Details on the flux measurements, data processing and quality control are provided in Burns et al. (2015).

## 2.3. The CARDAMOM Model-Data Fusion System

The CARbon DAta-MOdel FraMework (CARDAMOM; Bloom et al., 2016; Yin et al., 2020; Exbrayat et al., 2018; Smallman et al., 2017; Quetin et al., 2020; López-Blanco et al., 2019; Famiglietti et al., 2021; Bloom et al., 2020; amongst others) uses carbon cycle and meteorological observations to constrain carbon fluxes, states and process controls represented in the DALEC2 model of terrestrial C cycling (Williams et al., 2005; Bloom and Williams, 2015). Specifically, CARDAMOM uses a Bayesian model-data fusion approach to optimize DALEC2 time-invariant parameters (such as leaf traits, allocation and turnover times) and the "initial" C and $H_2O$ conditions (namely biomass, soil and water states at the start of the model simulation period).

The DALEC model (Williams et al., 2005; Rowland et al., 2014; Fox et al., 2009; Richardson et al., 2010; Famiglietti et al., 2021; Bloom & Williams, 2015; amongst others) is a box model of C pools connected via fluxes that has been used to evaluate terrestrial carbon cycle dynamics across a range of ecosystems and spatial scales. In all site, regional, and global applications, DALEC parameters are subject to very broad, but physically realistic, prior distributions, and independently estimated and constrained by available observations at each grid point. Here we use DALEC version 2 (DALEC2; Yin et al., 2020; Quetin et al., 2020; Bloom et al., 2020); gross and net carbon fluxes are determined as a function of 33 parameters, including 26 time-invariant parameters relating to allocation, turnover times, plant traits, respiration climate sensitivities, water-use efficiency and GPP sensitivity to soil moisture, and 7 parameters describing the initial conditions of live biomass pools (live biomass C, dead organic C and plant-available $H_2O$). Within DALEC2, GPP estimates are generated in the aggregated canopy model (ACM, Williams et al., 1997); the ACM is derived from simple functional relationships with environmental and plant structural and biochemical information (Williams et al., 1997), that are produced from a sensitivity analysis of GPP estimates from the more comprehensive SPA land surface model scheme (Williams et al., 1996, Williams et al., 2001). ACM GPP

estimates are contingent on plant structural and biochemical variables (including LAI, foliar nitrogen and nitrogen-
use efficiency) and meteorological forcing (total daily irradiance, maximum and minimum daily air temperature, day
length, atmospheric $CO_2$ concentration). In DALEC2, water limitation on ACM is prescribed as a linear response to
soil water deficit (Bloom et al., 2020). For more details on the model-data fusion methodology and CARDAMOM
ensembles, we refer the reader to Appendix A. For a comprehensive overview of the DALEC2 model, we refer the
reader to Bloom et al. (2020) and references therein.

### 2.4. Experiment Design

In order to develop model experiments that could reliably evaluate temperature-GPP interactions, we first
examine the observed environmental controls on tower-derived GPP. We focus on GPP during spring, defined here
as the period from March-May, which encompasses the climatological onset of GPP and transition from dormant
winter conditions to peak summer conditions (Fig 1a). Mean spring GPP exhibits large interannual variability (IAV)
with both a small decreasing trend from 2000-2010 (-0.02 g C m$^{-2}$ day$^{-1}$ per year) and increasing trend from 2010-
2018 (0.04 g C m$^{-2}$ day$^{-1}$ per year) (Fig. 1b). Comparison to tower observed temperature data (Fig. 1b and Fig. 2)
shows that spring GPP is positively correlated to mean spring air temperature (Pearson's linear r = 0.89, p =
0.000004) and summer (June-September) air temperature (r = 0.10, p = 0.70, Fig. S1a). Mean winter (December-
February) precipitation also has a positive correlation with spring GPP, (r = 0.07, p = 0.77, Fig. S1b), but it is much
smaller than spring temperature. At interannual timescales, mean annual GPP shows a small increasing trend
(0.0072 g C m$^{-2}$ day$^{-1}$ per year) over the time period (Fig. S2), and largest correlation with winter (December –
February) precipitation (Pearson's linear r = 0.63, p = 0.003, Fig. S3d) and shortwave irradiance (r = -0.30, p = 0.22,
Fig. S3f). In contrast, spring temperature shows little correlation with mean annual GPP (r = -0.02, p = 0.92, Fig.
S3c). It appears that winter precipitation and total irradiance are the dominant drivers in annual productivity, both of
which are correlated, while spring temperature show a first order effect in driving spring GPP.

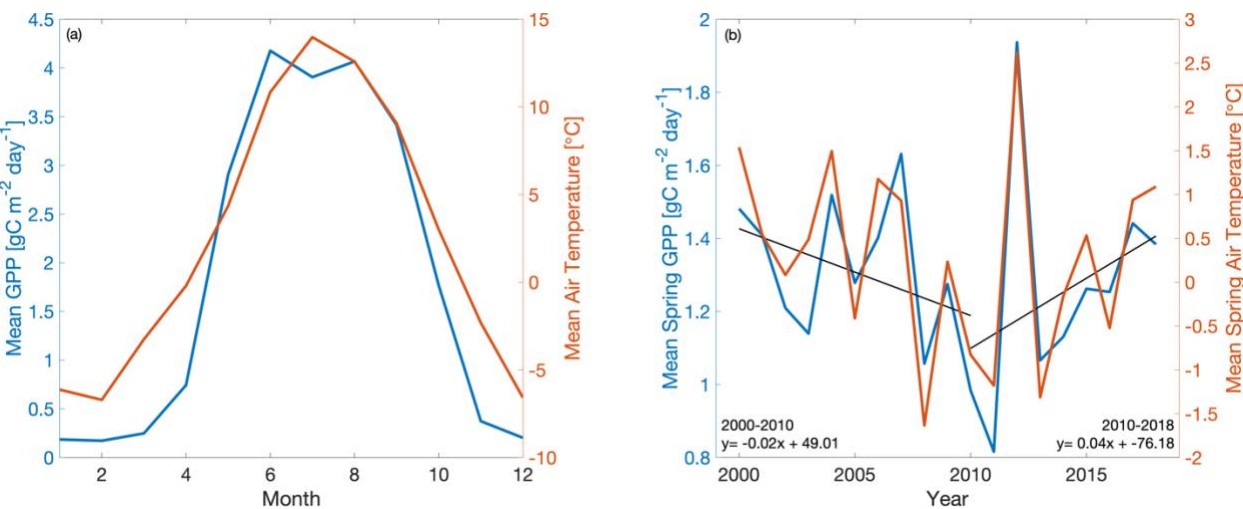

**Figure 1.** Time series of (a) mean monthly GPP (blue) and air temperature (orange) and (b) mean spring (March-May) GPP and
air temperature at Niwot Ridge (US-NR1) from 2000-2018. GPP data are derived using a nighttime partitioning technique based
on tower observations of NEE and air temperature.

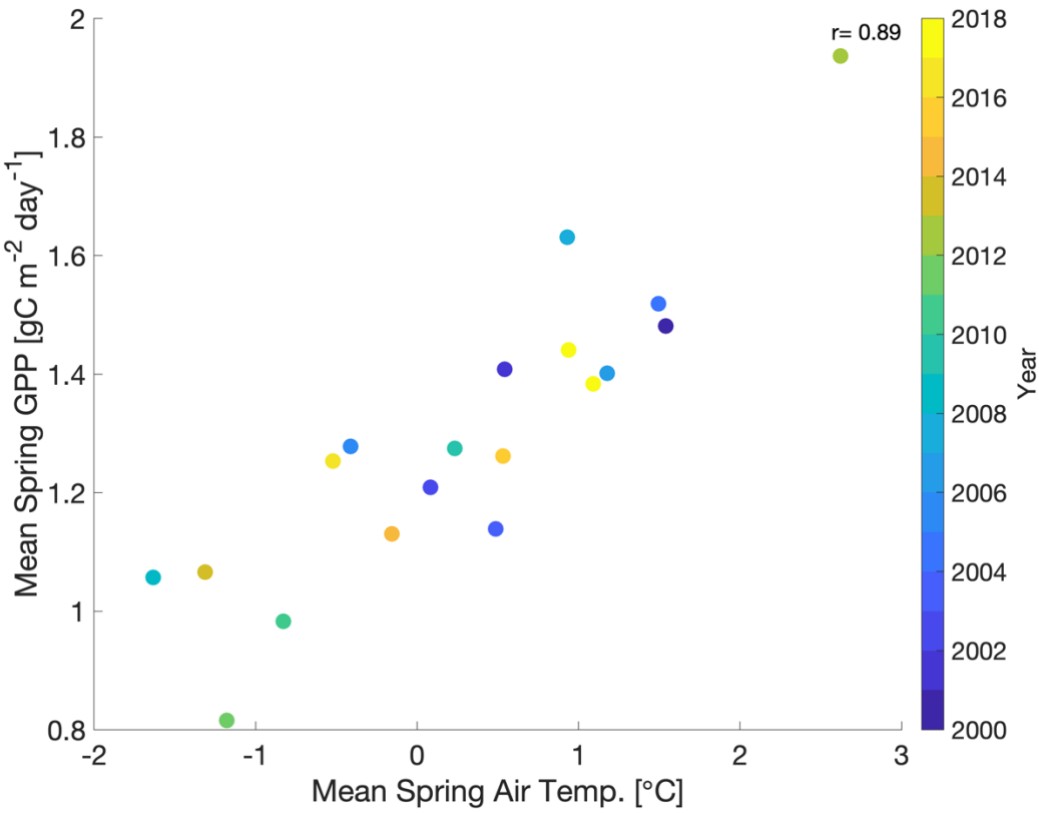

174

**Figure 2.** Scatterplot of mean spring (March-May) GPP with mean spring air temperature with the color bar showing the
corresponding year (2000-2018). 'r' is Pearson's correlation coefficient.

We also find that cold temperature has an important limitation on seasonal GPP at US-NR1.  The seasonal

cycle of GPP shows peak productivity in early summer (around June) and falling to near-zero values by early winter
(November), continuing through late winter (February-March).  Comparison of monthly GPP and minimum,
maximum, and mean monthly air temperature shows an initiation of photosynthesis at monthly maximum air
temperature above 0 °C (Fig. 3a) and monthly minimum air temperature above -5 °C (Fig. 3b).  The strong
dependence of monthly GPP on temperature is consistent with previous findings that temperature is an important
driver of spring onset and seasonal variability of GPP in evergreen forests (e.g., Pierrat et al., 2021; Parazoo et al.,
2018; Euskirchen et al., 2014; Arneth et al., 2006).  As temperature falls in winter dormant plants, productivity
becomes negligible.  Productivity is triggered again when spring air temperature becomes warm enough to thaw
stems, trigger xylem flow and promote access to soil moisture (e.g., Pierrat et al., 2021; Bowling et al., 2018; Ishida
et al., 2001).  Due to this observed dependence of GPP on temperature at US-NR1, we focus our analysis
specifically on spring GPP, where we hypothesize that cold temperature is the dominant control on spring GPP
variability.

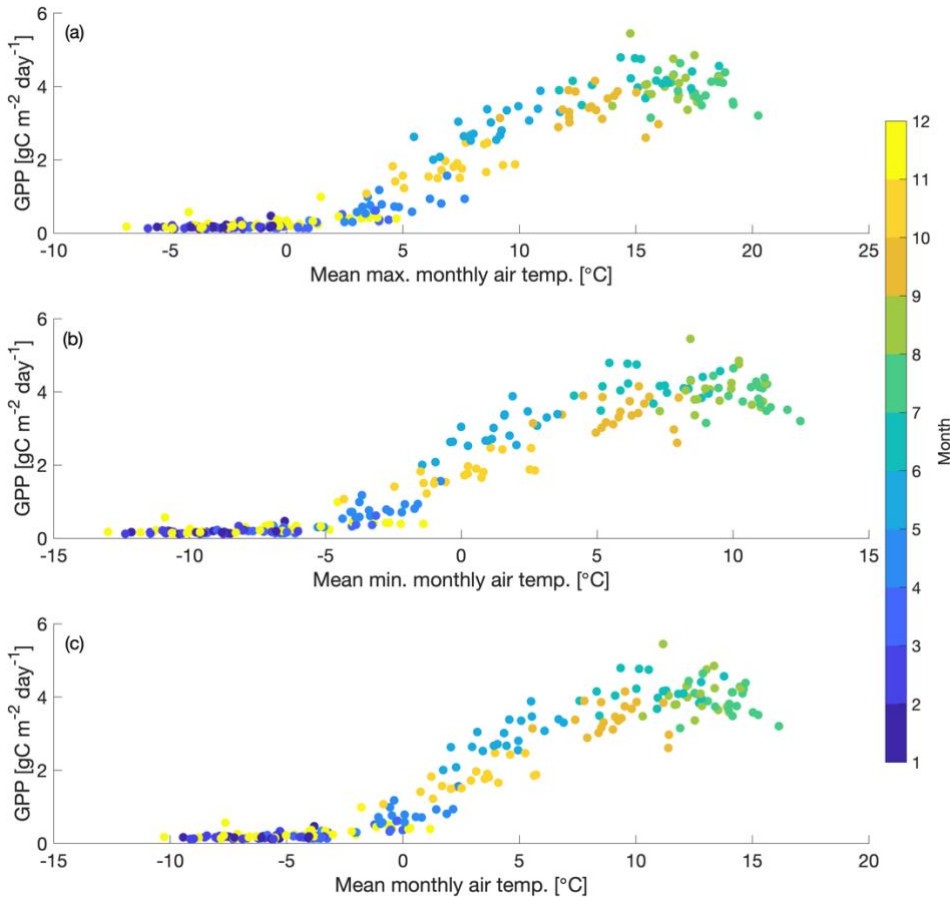

**Figure 3.** Scatter plot of mean monthly GPP vs. a.) mean maximum air temperature, b.) mean minimum air temperature and c.) mean air temperature for 2000-2018. Dots are colored with the corresponding month.

In the baseline version of CARDAMOM, seasonal GPP in DALEC2 is limited primarily by incoming shortwave radiation. This light-focused limitation works well for deciduous forests where spring temperature and sunlight are correlated, as well as high latitude regions where sunlight is limited. However, for reasons discussed above, this method fails in evergreen forests such as Niwot Ridge whose green canopies are exposed to high sunlight and below-freezing temperature in spring. As temperature increases, evergreen stems slowly thaw, which enables the trees to access available soil moisture and slowly reactivate their carbon and water exchange processes (Mayr et al., 2014; Bowling et al., 2018). Temperature also impacts the reactivation of photosynthetic activity after winter dormancy (Öquist and Huner, 2003; Tanja et al., 2003). For example, fluctuating temperature in the spring has been shown to limit and sometimes reverse the activation of biochemical processes needed for photosynthesis recovery (Ensminger et al., 2004). Exposure to cold temperature, when combined with increased irradiance in the spring, can also damage evergreen trees (Öquist and Huner, 2003; Yang et al., 2020), therefore disrupting $CO_2$ assimilation. Previous studies have captured these cold temperature impacts at Niwot Ridge and other evergreen sites. For example, variations in photosynthetic pigments have been tied to seasonal temperature at Niwot Ridge (Magney et al., 2019). Pierrat et al. (2021) identified an increase in plant water flow (measured via changes in diurnal stem radius) and a change in carotenoid-chlorophyll ratios as temperature increases. The activation of water flow in the

evergreen trees, combined with the pigment changes to absorb more sunlight, allows for the recovery of
photosynthesis in the spring.
To represent the integrated impact of the cold weather processes, here we implement a cold temperature
scaling factor ($g$) in DALEC2.  This scaling factor is developed by analyzing the relationship between monthly
minimum & maximum air temperature with tower-derived monthly GPP, where
$If: T_{min}(t) < T_0 : g = 0$  (1)
$If : T_{min}(t) > T_g : g = 1$
$Else: g(t) = \frac{(T_{min}(t)-T_0)}{(T_g-T_0)}$
$GPP_{cold}(t) = GPP(t) * g(t)$  (2)
$T_{min}(t)$ is the observed minimum air temperature at Niwot Ridge at time $t$, GPP($t$) is the nominal ACM-based
DALEC2 GPP estimate (see section 2.3) and $GPP_{cold}$ is the corresponding cold temperature GPP estimate. Equation
(2) may represent the integrated effect of all cold weather biophysical limitations, including processes such as the
impact of cold weather on plant hydraulics, and changes to carotenoid-chlorophyll ratios.  We also theorize that our
temperature scaling factor partially captures soil moisture disruptions due to changing soil temperature.  The
temperature thresholds in Eq. (1) may account for the connection between air temperature and soil temperature, with
initial and full soil thawing temperature potentially mirroring the photosynthesis shutdown and initiation air
temperature.  CARDAMOM does not currently have explicit representations of soil moisture stress due to soil
freezing.  Therefore, soil freezing stress and other biophysical processes impacted by cold temperature may be
approximated by this cold temperature scaling factor added to CARDAMOM. The temperature thresholds for
photosynthesis shutdown (referred to as $T_0$) and initiation (referred to as $T_g$) are added as model parameters in
DALEC2, bringing the total number of parameters to 35.  These 35 DALEC parameters are simultaneously
optimized in CARDAMOM. The CARDAMOM Bayesian-inference probability distributions (see Appendix A) for
the $T_0$ (-7.1 $\pm$ 1.1 °C) and $T_g$ (6.0 $\pm$ 2.6 °C) parameters used to define the cold temperature limitation are plotted in
Fig S4. We refer to the cold temperature constrained version of DALEC2 (within CARDAMOM) as DALEC2cold.
The baseline (DALEC2) and cold temperature (DALEC2cold) versions of the model are run for the 2000-
2018 period using tower observed, gap-filled, monthly meteorological (MET) drivers (including minimum and
maximum air temperature, shortwave radiation, vapor pressure deficit, and precipitation).  We conduct four
experiments, summarized in Table 1: experiments using DALEC2 and DALEC2cold within CARDAMOM, where
19 years of GPP data are assimilated (referred to as CARD and CARDcold), and a corresponding pair of
experiments where only the first decade of data (2000-2009) is assimilated (referred to as CARD-Half and
CARDcold-Half) and the second decade of data (2010-2019) is withheld for validation, as a train-test scenario.  All
months of GPP data are assimilated into the model, however our analysis focuses on the constraints on spring
(March-May) GPP. These four experiments serve to evaluate the sensitivity of modeled GPP at Niwot Ridge to cold
temperature limitation and parameter optimization. Specifically, the objective of experiments "CARD" and
"CARDcold" is to determine whether the cold temperature scaling factor improves the representation of spring GPP
variability across the 2000-2018 period; the objective of experiments "CARD-Half" and "CARDcold-Half" is to
cross-validate the predictive skill of CARDcold by assessing whether the addition of a cold temperature scaling
factor, informed by a subset of GPP data, can improve prediction of a withheld subset of GPP data.

**Table 1.** Summary of CARDAMOM modeling experiments to determine sensitivity of seasonal and interannual spring GPP
variability to cold temperature limitation (CARD vs CARDCold) and the ability to perform outside training window (Half).

| Experiment Name | Met. Drivers | Time Period | GPP assimilation | Time period considered in assimilation | Uncertainties in GPP | Cold Temp. Limitation |
|---|---|---|---|---|---|---|
| CARD | yes | 2000-2018 | yes | 2000-2018 | 20% | No |
| CARD-Half | yes | 2000-2018 | yes | 2000-2009 | 20% | No |
| CARDCold | yes | 2000-2018 | yes | 2000-2018 | 20% | Yes |
| CARDCold-Half | yes | 2000-2018 | yes | 2000-2009 | 20% | Yes |


### 2.5. Comparison to Terrestrial Biosphere Model Ensemble

A recent model intercomparison study provides an ideal benchmark for evaluating CARDAMOM

simulations (section 2.4). Parazoo et al. (2020) conducted an experiment in which an ensemble of state-of-the-art
terrestrial biosphere models (TBMs) were forced by the same observed meteorology at Niwot Ridge from 2000-2018,
but with differences in spin-up, land surface characteristics, and parameter tuning. The TBMs are designed to simulate
the exchanges of carbon, water, and energy between the biosphere and atmosphere, from global to local scales
depending on inputs from meteorological forcing, soil texture, and plant functional type (PFT). The experiment was
designed primarily to evaluate simulations of solar induced fluorescence (SIF) and GPP, the latter of which we focus
on here. We refer the reader to Parazoo et al. (2020) for a more complete description of models, within-model
experiments, and between-model differences.

The most important model differences worth noting here include the representation of stomatal conductance,

canopy absorption of incoming radiation, and limiting factors for photosynthesis. We analyze a subset of the models
which were run for multiple years, including SiB3 and SiB4 (Simple Biosphere model versions 3 and 4, respectively),
ORCHIDEE (Organizing Carbon and Hydrology in Dynamic Ecosystems), BEPS (Boreal Ecosystems Productivity
Simulator), and CLM4.5 and CLM5.0 (Community Land Model Versions 4.5 and 5.0, respectively). We also analyze
within-model experiments in SiB3 and ORCHIDEE to isolate effects related to prescription of leaf area index (LAI;
monthly varying in SiB3-exp1, fixed at 4.0 $m^2/m^2$ in SiB3-exp2), temperature and water stress (ORCHIDEE-exp1
includes temperature stress; ORCHIDEE-exp2 accounts for temperature and water stress), and data assimilation
(ORCHIDEE-exp3, in which a subset of model parameters controlling photosynthesis and phenology are optimized
against global OCO-2 SIF data, Bacour et al., 2019).  Most of the TBM model experiments were run with default
parameters (BEPS, CLM50, SiB3, SiB4, ORCHIDEE-exp1 and exp2). The other experiments were optimized in the
following ways: either a) parameters were hand-tuned based on the US-NR1 data (CLM45) or b) the parameters were
optimized using OCO-2 SIF data (ORCHIDEE-exp3). For more details on the parameterization of the TBM-SIF
experiments, we refer the reader to Parazoo et al. (2020). The use of these models provides insight into the spread in
model structures and the use of their default parameters. Finally, we note that not all model simulations span the entire
observed record (2000-2018). While our analysis focuses on the long-term record from 2000-2018, we provide
multiple comparisons to ensure consistency of time period: (1) IAV from 2001-2018 for SiB3, SiB4, ORCHIDEE,
and CLM4.5; (2) IAV from 2012-2018 for SiB3, SiB4, ORCHIDEE, CLM4.5, and CLM5.0, and (3) seasonal
variability from 2015-2018 for all models. We refer to the ensemble of models and within model experiments
collectively as TBM-MIP.

### 283    3.   Results & Discussion

### 284    3.1. Evaluation of CARDAMOM 2000–2018 GPP

When the 19 years of tower-derived GPP data are assimilated into both versions of the model, the mean
seasonal cycle is accurately replicated (Fig. 4). The Pearson's r values for CARD (Fig. 4a) and CARDcold (Fig. 4b)
are almost equal (r = 1.0 and 0.99) with minimal increases in root mean square error (RMSE) and mean bias error
(MBE) for CARDcold (RMSE = 0.24 g C m$^{-2}$ day$^{-1}$ and 0.23 g C m$^{-2}$ day$^{-1}$, MBE = 0.06 g C m$^{-2}$ day$^{-1}$ and 0.19 g C
m$^{-2}$ day$^{-1}$ for CARD and CARDcold, respectively). Assimilating only the first decade of GPP data (Half
experiments) does not drastically alter model performance (Fig. S5), with only slight changes in RMSE and MBE
($\Delta$RMSE = 0.008 g C m$^{-2}$ day$^{-1}$, $\Delta$MBE = 0.03 g C m$^{-2}$ day$^{-1}$ for CARD-Half, $\Delta$RMSE = -0.003 g C m$^{-2}$ day$^{-1}$,
$\Delta$MBE = 0.02 g C m$^{-2}$ day$^{-1}$ for CARDcold-Half).

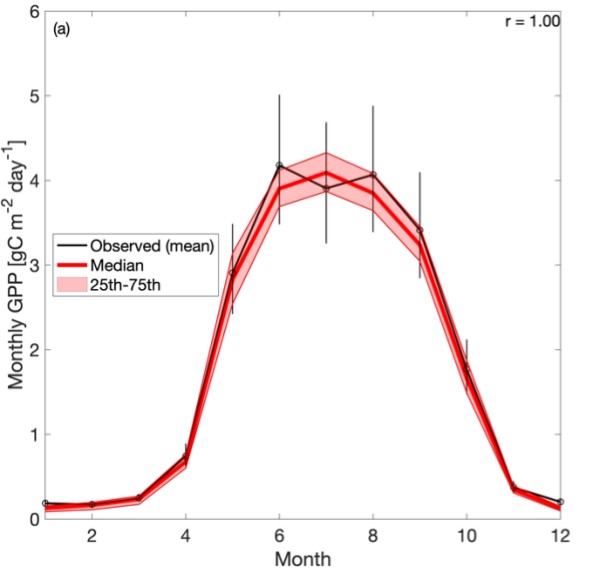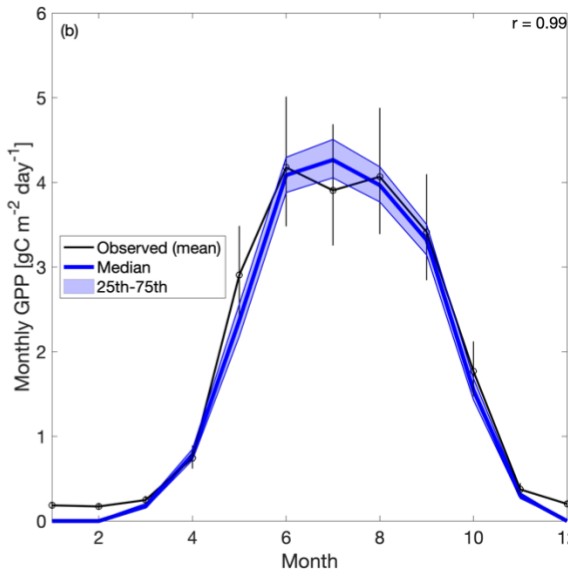


**Figure 4.** Tower-derived average monthly GPP (black line) and modeled GPP seasonal cycles at US-NR1 for 2000-2018, for a.)
CARD and b.) CARDcold experiments. The half-assimilation experiments (CARD-Half and CARDcold-Half) can be found in
the supplement (Fig S5). Model outputs include the median value of each experiment (bold color line) with the 25th-75th
percentiles of the ensembles (shaded area). The median is plotted instead of the mean to avoid impact of outlier ensemble
members (N = 4000). Error bars = tower-derived GPP multiplied/divided by exp(sqrt(log(2)^2*n)/n), n=# of years in average (n
= 19). 'r' is the Pearson's coefficient.

The cold experiments exhibit an improved fit to the observed IAV in spring productivity (Fig. 5), relative to
CARD, (r = 0.47, std = 0.03 g C m$^{-2}$ day$^{-1}$ for CARD; r = 0.88, std = 0.27 g C m$^{-2}$ day$^{-1}$ for CARDcold). CARDcold
also has slightly reduced RMSE (-0.01 g C m$^{-2}$ day$^{-1}$) and larger MBE (0.13 g C m$^{-2}$ day$^{-1}$). Similar to the seasonal
cycle analysis, the assimilation of only the first decade of GPP data (Half experiments) has minimal impact on
model performance ($\Delta$RMSE = 0.007 g C m$^{-2}$ day$^{-1}$, $\Delta$MBE = 0.06 g C m$^{-2}$ day$^{-1}$ for CARD-Half, and $\Delta$RMSE =
0.02 g C m$^{-2}$ day$^{-1}$, $\Delta$MBE = 0.02 g C m$^{-2}$ day$^{-1}$ for CARDcold-Half). We find less agreement between modeled and
tower-derived GPP IAV in summer for both CARD and CARDcold (CARD r = 0.32, std = 0.11 g C m$^{-2}$ day$^{-1}$;
CARDcold r = 0.05, std = 0.10 g C m$^{-2}$ day$^{-1}$; Fig. S6). While there is little variation in RMSE between the half and
full-assimilation experiments, RMSE is larger for summer than spring GPP (average RMSE = 0.23 g C m$^{-2}$ day$^{-1}$ for
spring model outputs, average RMSE = 0.35 g C m$^{-2}$ day$^{-1}$ for summer model outputs). Model agreement is further
reduced when considering annual average GPP (Fig. S7, Table S2). Although the cold temperature limitation
improves IAV slightly, it is still small compared to observed variability (mean annual std = 0.14 g C m$^{-2}$ day$^{-1}$).
Correlations to tower-derived GPP at the annual scale are small for both CARD and CARDcold (r = 0.19 and r =
0.22, Fig. S7a-b). Overall, the cold temperature limitation substantially improves agreement between the model and
tower-derived spring GPP, with slight reductions in performance for summer and annual GPP.

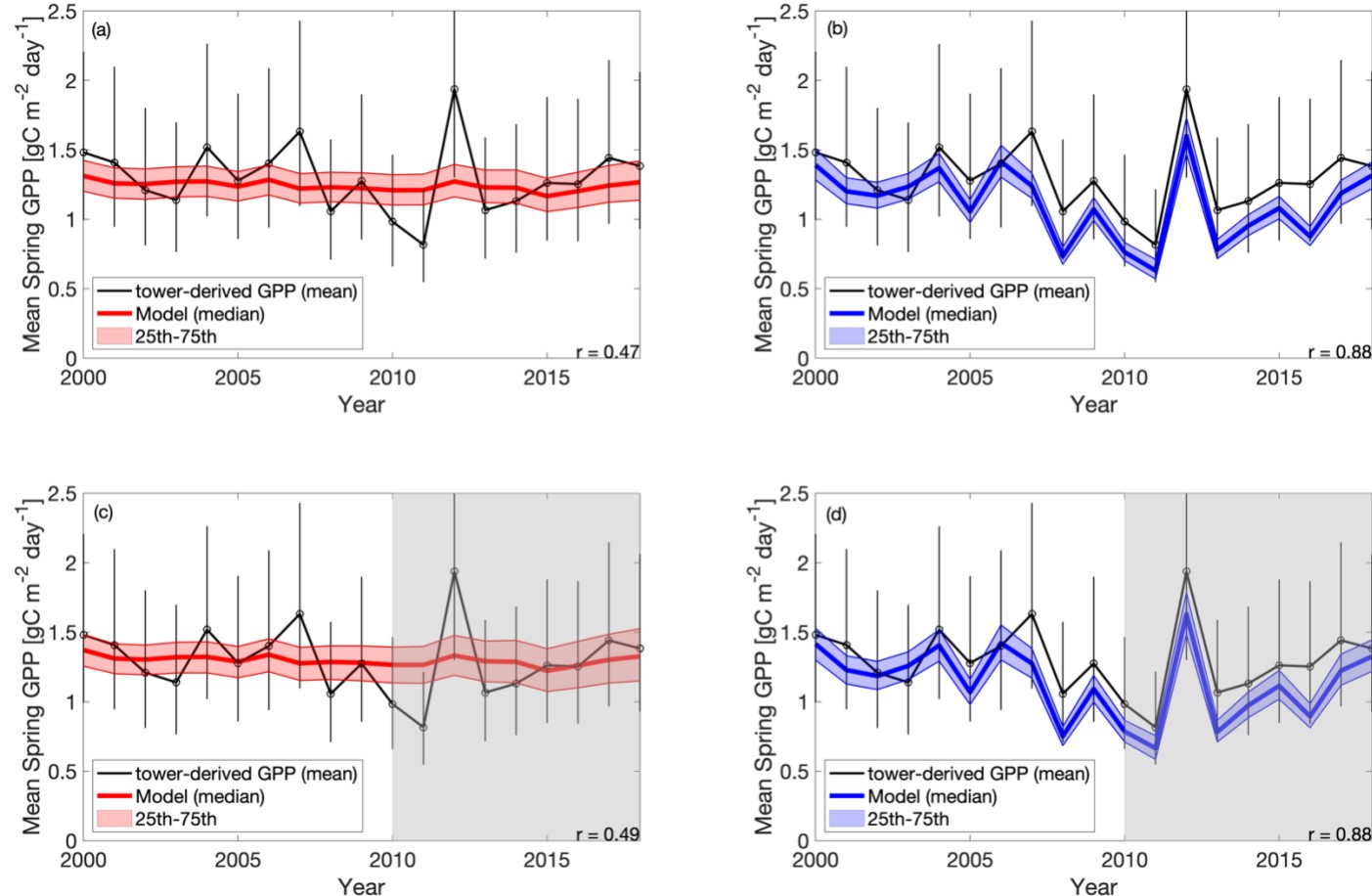

**Figure 5.** Tower-derived (black line) mean spring (March-May) GPP with model interquartile range (shaded area) and median
(bold color line) spring GPP outputs for a.) CARD, b.) CARDcold, c.) CARD-Half, and d.) CARDcold-Half experiments. The
grey regions indicate no data assimilation (i.e. testing window). Model experiments are the same as in Figure 4. Uncertainty =
exp(sqrt(log(2)^2*n)/n), n=# of months in average (n = 3).

The standard deviation in tower-derived mean spring GPP (March-May) is approximately 0.25 g C m$^{-2}$ day$^{-}$

$^{1}$. The addition of the cold temperature limitation improves the model's ability to match the IAV of mean spring
GPP (Fig. 6a-b). An examination of all modeled scenarios for CARD and CARDcold (i.e. all 4000 DALEC2
simulations), shows that the cold temperature limitation produces spring IAV values much closer to what is
observed in the tower-derived GPP data. Only 0.3% of CARD ensembles produces mean spring IAV values within
20% of the tower-derived spring GPP standard deviation (0.25 ± 0.05 g C m$^{-2}$ day$^{-1}$), whereas 69% of CARDcold
ensembles have standard deviation values within the same range. Interestingly, assimilating only the first ten years
of GPP data (Half experiments, Fig. 6b) slightly increases the number of ensemble members with standard
deviations within the mentioned range for both CARD-Half (2.4%) and CARDcold-Half (70%). It is promising to
see that despite not assimilating the 2010-2018 GPP data into the model, CARDcold-Half is still able to match
average spring IAV of the full data record.

We also consider the IAV in spring GPP for just the second half of the data record (2010-2018). IAV of

tower-derived spring GPP increases slightly in 2010-2018 (0.32 g C m$^{-2}$ day$^{-1}$). Once again, the cold temperature
limitation enables CARDAMOM to match spring GPP IAV (Fig. 6c-d). 0.03% of CARD ensembles produce mean
spring IAV values within 20% of the tower-derived spring GPP standard deviation for the 2010-2018 period (0.32 ±
0.06 g C m$^{-2}$ day$^{-1}$), whereas 76% of CARDcold ensembles have standard deviation values within the same range.
For the Half experiments, 0.6% of CARD and 75% of CARDcold ensembles have IAV values within 20% of the
standard deviation for 2010-2018. This improvement in matching IAV is also observed when considering mean
annual GPP (Fig. S8), but is much smaller than the improvements made for spring GPP. Overall, CARDcold
produces a less biased distribution of IAV values (relative to both assimilated and withheld observations), whereas
CARD is more skewed towards smaller IAVs, which indicates that the cold temperature limitation enables a
mechanistic and statistical improvement in capturing the interannual variability of spring GPP.

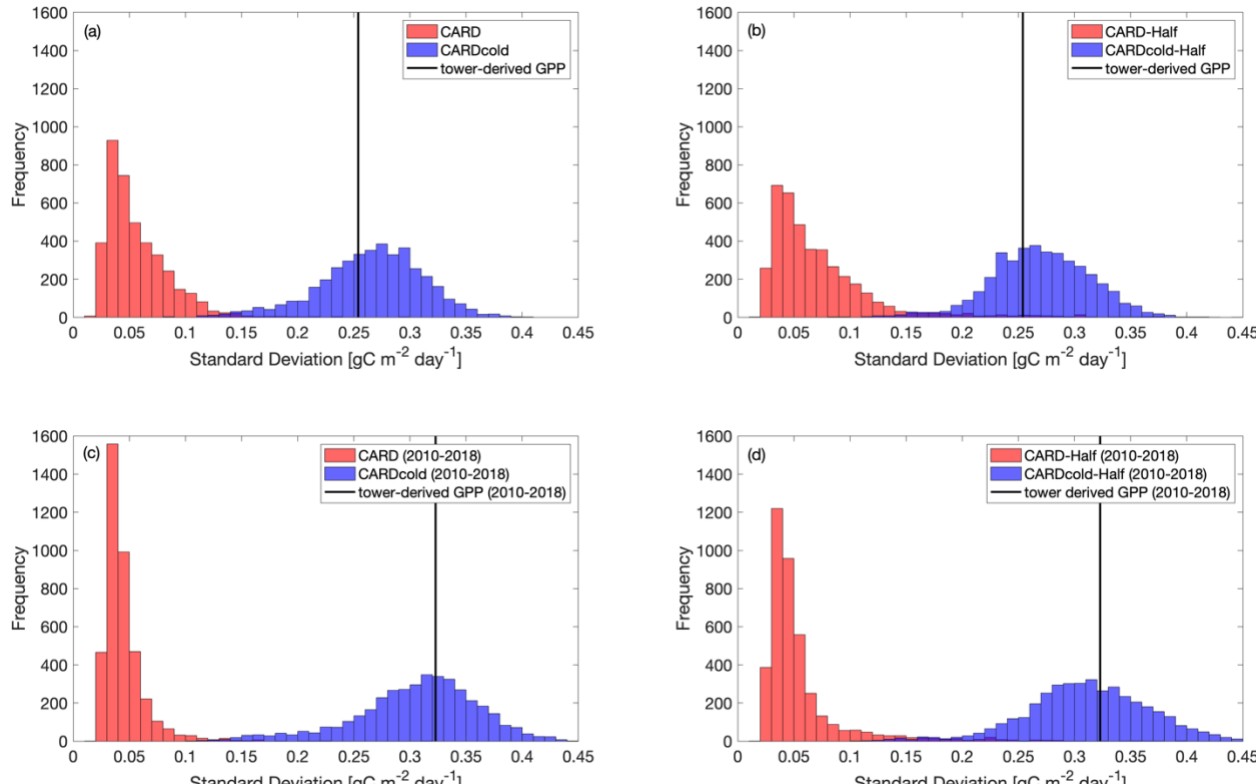

**Figure 6.** Histograms comparing standard deviation in mean spring GPP across all ensembles (N=4000) for CARD (red bars) and
CARDcold (blue bars) experiments with a.) full assimilation, b.) half assimilation, c.) full assimilation for the second decade
(2010-2018), and d.) half assimilation for the second decade (2010-2018). Black line indicates standard deviation in tower-
derived mean spring GPP (std = 0.25 gC m$^{-2}$ day$^{-1}$ for full period (a-b), std = 0.32 gC m$^{-2}$ day$^{-1}$ for 2010-2018 (c-d)).

**3.2. Temperature controls on springtime GPP**
The added value of the DALEC2 cold temperature limitation for modeling mean spring (March-May) GPP
is logically due to large fluctuations in spring temperature at Niwot Ridge.  The cold temperature limitation allows
DALEC2-CARDAMOM to match the IAV of spring tower-derived GPP closely.  Furthermore, the cold temperature
limitation enables the model to match tower spring IAV in the second half of the time period (2010-2018) when only
the first ten years of GPP data are assimilated (2000-2009).  This indicates that the cold temperature limitation is
able to estimate spring GPP outside of its training window and could be useful at other sites where data availability
is limited. Future work will include evaluating the cold temperature limitation at other sites to ensure that it is
applicable beyond Niwot Ridge, for example using forecast skill metrics proposed by Famiglietti et al. (2021).

Temperature-induced spring onset of GPP is driven by two general processes: (1) initiation of bud burst

and leaf expansion leading to increasing LAI, and/or (2) initiation of photosynthetic activity (photosynthetic
efficiency i.e., GPP per unit of LAI) due to temperature-induced changes in plant hydraulics (Ishida et al., 2001;
Pierrat et al., 2021) or kinetics of the photosynthetic machinery (e.g., Medlyn et al., 2002). In situ LAI
measurements suggest that the LAI at Niwot Ridge is relatively constant across the season, which is somewhat
expected given the dominant tree species at the site. Hence, the temperature-induced onset of GPP is likely due to
the latter process, increased photosynthetic efficiency, as supported by the measurements (Figs. 1-2), although small
changes in LAI are still feasible given uncertainties in the measurements. The inclusion of the cold temperature
limitation scaling factor in the model, a semi-empirical process, leads to a substantial improvement in model
representation of GPP at the site. Further development may also look to identify the relative roles of increased LAI
and increased photosynthetic efficiency at Niwot Ridge and other evergreen needleleaf sites, as changes in GPP can
lead to changes in carbon allocation to LAI, among other plant carbon pools.

Temperature is important in both the reactivation of photosynthetic activity in the spring and the wind

down of productivity in the fall (Flynn and Wolkovich, 2018; Stinziano and Way, 2017). Therefore, we anticipate
that the cold temperature scaling function may also improve our ability to model fall productivity. However, other
factors such as water availability and photoperiod must also be considered (Bauerle et al., 2012; Stinziano et al.,
2015). Future studies at Niwot Ridge and other sites should investigate the role of these factors (temperature, water,
photoperiod) in regulating fall GPP and how we can represent these processes in CARDAMOM.

With the inclusion of the cold temperature limitation on GPP and its application in CARDAMOM, we

provide a data-constrained estimate of the climate sensitivity of the Niwot Ridge forest to spring temperature.
Posterior estimates indicate that GPP is gradually inhibited below 6.0 °C ± 2.6 °C ($T_g$) and completely inhibited
below -7.1 °C ± 1.1 °C ($T_0$). The gradual limitation of GPP by temperature has been observed on hourly and daily
timescales in other cold-weather ecosystems, such as Alaskan conifers (Parazoo et al., 2018) and Canadian spruce
(Pierrat et al., 2021). This has been connected to the triggering of transpiration and water flow from xylem into
leaves (Ishida et al., 2001). However, both biotic (e.g., carotenoid/chlorophyll ratios) and abiotic (e.g., openness of
canopy) factors together regulate GPP response to meteorological forcings, and further process-oriented
investigations are required to resolve the emergent response of GPP to temperature. Furthermore, the use of
process-based models will be needed to disentangle the individual cold weather biophysical processes currently
represented in the scaling factor (Eq. 1-2). For now, this is a useful metric for climate-sensitivity of spring GPP, at
least in the absence of long-term adaptations. Furthermore, over the 19 year observation period investigated here the
use of a temporally constant $T_0$ and $T_g$ yields significantly improved GPP estimates, suggesting that much of the
variability can be attributed to climate-driven changes, not interannual variation in vegetation parameters. As
temperature continues to increase due to climate change (particularly in the early growing season), productivity at
US-NR1 could increase as a result and therefore increase carbon uptake, with productivity peaking earlier in the
year (e.g., Xu et al., 2016). However, these spring gains in GPP have been shown to not offset the losses of carbon
due to summer droughts (e.g., Buermann et al., 2013; Knowles et al., 2018). It is also unclear how the long-term
stress of increased temperature could affect forest productivity directly.

This study focuses on the relationship between temperature and GPP and its usefulness on model

predictions of spring GPP, but an important component that cannot be ignored is the confounding effect of water
availability on GPP. Future changes in winter precipitation are more uncertain, therefore limiting our ability to
analyze how precipitation changes will alter future productivity. While precipitation observations are analyzed to
discern any major connections between GPP and meteorological controls, an analysis of how precipitation affects
model predictability is not included in this study. The combined results, including the cold temperature limitation
and train-test data assimilation experiments, suggest that other factors besides spring temperature, most notably
winter and summer precipitation (Fig. S3) and resulting soil water limitation, also have important impacts on
summer GPP. We therefore highlight the need to jointly resolve springtime temperature limitation in conjunction
with water stress limitations in future efforts to understand the integrated role of environmental forcings on
interannual GPP variability. Furthermore, this analysis does not consider how winter precipitation as snowfall
versus rainfall affects productivity, or how resulting changes to winter snowpack could alter productivity long-term.
Since annual average GPP appears to be more dependent on winter precipitation/snowpack (Pearson's linear r =
0.63, Fig. S3a), future work will include improving model predictability of late season productivity and quantifying
temperature-water effects on carbon uptake. The definition of the seasons could also alter the connections drawn
between seasonal temperature, precipitation and productivity.
**3.3. Model intercomparison and implications for GPP models**

Here, we evaluate DALEC2-CARDAMOM against mean spring GPP estimates from TBM-MIP models

(Section 2.5 and Parazoo et al. 2020). It is important to remind the reader that the CARDAMOM runs have a
significant advantage over the TBM-MIP models in this analysis, as CARDAMOM is trained on US-NR1 GPP data.
While TBM-MIP models use tower-observed meteorological inputs, prescribe tower-specific and time-invariant
structural properties such as LAI observed at US-NR1 (SiB3-exp2 and CLM4.5), and use data assimilation of global
remote sensing data to constrain globally representative plant functional types (ORCHIDEE-exp3), they are not
directly constrained by time-varying carbon fluxes at the tower. As such, we emphasize that our model comparison
is not a strict assessment of performance, but rather an attempt to learn how model simulation of GPP at an
evergreen needleleaf site can be improved.

There is a wide range in performance of TBM-MIPs in representing the magnitude and IAV of tower-

derived spring GPP (Figure 7a). Pearson's r correlations range from 0.25 to 0.82 (mean r = 0.6, Table 2) from 2001-
2018, with the same models showing slightly improved performance over the second decade (mean r = 0.73 from
2012-2018). ORCHIDEE-exp1 and CLM4.5 show consistently high performance over all three periods analyzed,
with CLM5.0 excelling from 2012-2018, and BEPS from 2015-2018 (Table S1). CLM4.5 also shows the smallest
mean bias of the TBM-MIP models (RMSE ~ 0.35), and high agreement in the magnitude of spring GPP variability
(1-sigma standard deviation = 0.21 g C m$^{-2}$ day$^{-1}$ for CLM4.5, vs 0.25 g C m$^{-2}$ day$^{-1}$ observed). While
acknowledging the advantage of data assimilation, it is promising to see that CARDAMOM (with the addition of the
cold temperature limitation) is able to perform comparably to the TBM-MIP models. In particular, CARDcold is
well correlated in the direction (r = 0.88) and magnitude (1-sigma ~0.26) of interannual variability, as well as overall
magnitude of spring GPP (low RMSE and MBE).

The range of performance across within-model experiments reveals important processes, and uncertainty of

process representation, in driving the magnitude and variability of spring GPP. For example, the ORCHIDEE data
assimilation experiment (exp3) shows consistently and substantially lower overall correlation (e.g., r = 0.59 from
2001-2018) than corresponding free running experiments (exp 1 and 2, r = 0.78-0.82), but has reduced RMSE and
MBE (RMSE = 0.63 g C m$^{-2}$ day$^{-1}$ vs 1-1.14 g C m$^{-2}$ day$^{-1}$). Likewise in SiB3, prescribing an empirically-based but
fixed-in-time LAI of 4.0 m$^2$/m$^2$ (exp2) reduces mean bias, but degrades variability (r = 0.25) compared to time-
variable LAI (exp1) prescribed from satellite data (r = 0.50).


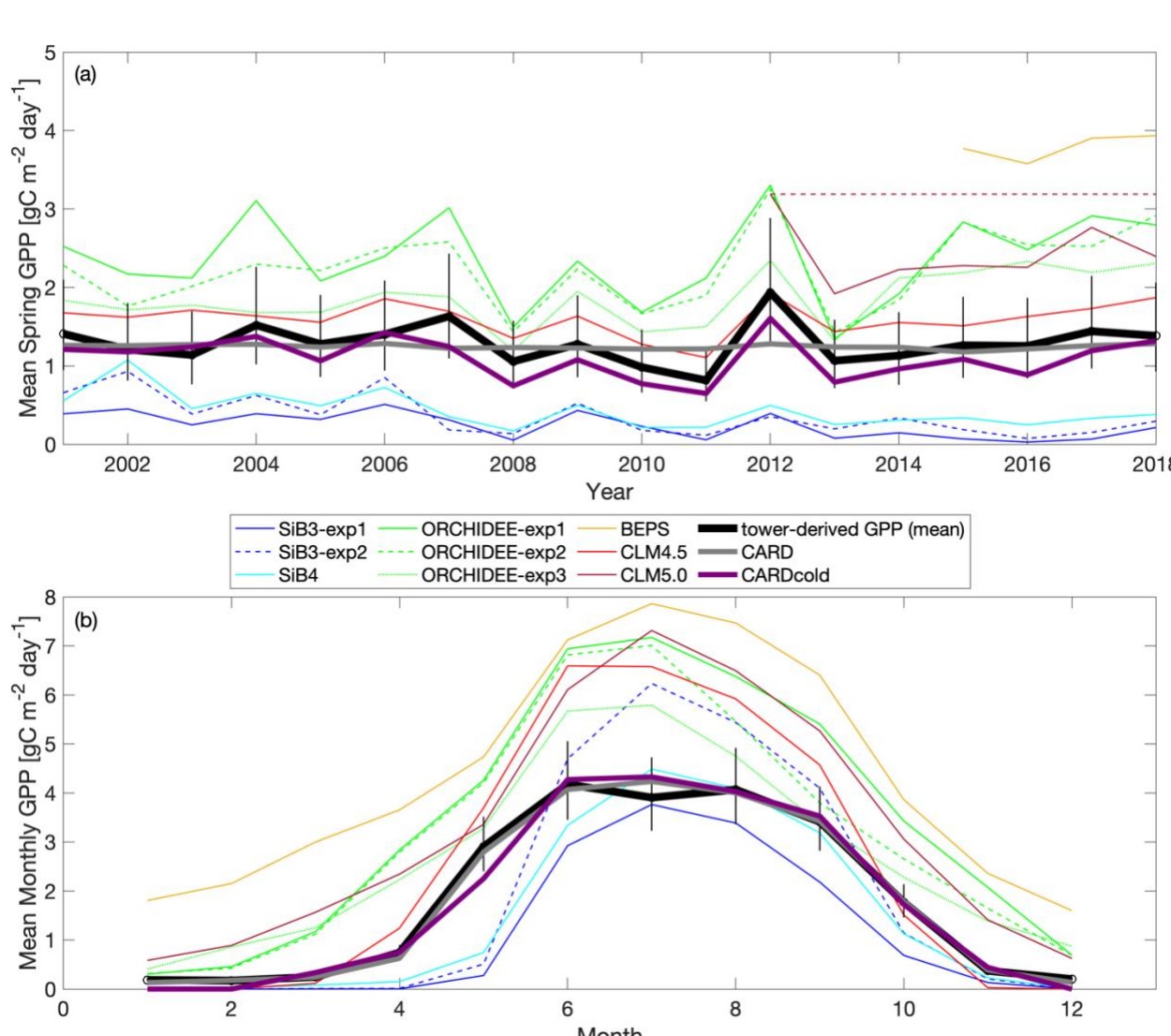



**Figure 7.** Comparison of TBM-MIP models to CARD and CARDcold experiments for a.) mean spring GPP for 2000-2018 and
b.) monthly GPP from 2015-2018. Note that fill values are ignored when calculating mean annual values for TBM-MIP
experiments. Uncertainty = $\exp(\sqrt{\log(2)^2 \cdot n)/n})$, where n = # years in average (n = 19).

There is also large variability in the modeled seasonal cycle (Fig. 7b) and mean annual GPP (Fig. S9). For

mean annual GPP estimates, Pearson's r values are reduced for all models (Table S2). Once again, ORCHIDEE-
exp2 and ORCHIDEE-exp3 stand out with some of the higher correlations (r = 0.60 and r = 0.64) and p-values
below 5% significance level. Furthermore, ORCHIDEE-exp3 (temperature stress with SIF data assimilation) has
the lowest RMSE and MBE of the model set. SiB3-exp2 (fixed LAI) has a standard deviation closest to
"observations" (0.14 gC $m^{-2}$ day$^{-1}$), and the smallest RMSE and MBE of the TBM models.

Most TBM-MIP models capture the shape of the seasonal cycle at Niwot Ridge. For the 2015-2018 period,

all models have Pearson's r values larger than 0.91, with p-values much smaller than a 5% significance level (Table
S3). With the help of data assimilation, CARDcold accurately captures the seasonal cycle at Niwot Ridge with
reduced error (RMSE = 0.22 g C $m^{-2}$ day$^{-1}$, MBE = 0.07 g C $m^{-2}$ day$^{-1}$), and data assimilation experiments in
ORCHIDEE-exp3 show reduced bias relative to free running experiments. The cold temperature limitation has little
impact on the modeled mean seasonal cycle or mean annual GPP estimates in CARDAMOM, and appears to be
most valuable for improving spring GPP variability.

**Table 2.** Pearson's linear r, R-squared, p-value, standard deviation, root mean square error (RMSE), and mean bias error (MBE)
for TBM-MIP and all CARDAMOM experiments to Niwot Ridge tower-derived mean spring (March-May) GPP. Open values
reflect statistics for the 2001-2018 period, while values in parentheses represent the 2012-2018 period. All relevant statistics are
calculated at 5% significance level. *BEPs statistics are not included in this table as this model only has GPP estimates for 2015-
468 2018.

| model | r-value | R-squared | p-value (α = 0.05) | RMSE (gC $m^{-2}$ d$^{-1}$) | MBE (gC $m^{-2}$ d$^{-1}$) | standard deviation (gC $m^{-2}$ d$^{-1}$) |
|---|---|---|---|---|---|---|
| CARD-Half | 0.47 (0.55) | 0.22 (0.30) | 0.05 (0.20) | 0.24 (0.26) | -0.005 (0.06) | 0.03 (0.04) |
| CARD | 0.45 (0.57) | 0.20 (0.33) | 0.06 (0.18) | 0.24 (0.28) | 0.05 (0.12) | 0.03 (0.04) |
| CARDcold-Half | 0.88 (0.93) | 0.77 (0.86) | 0.00 (0.002) | 0.21 (0.24) | 0.17 (0.22) | 0.26 (0.29) |
| CARDcold | 0.87 (0.93) | 0.76 (0.87) | 0.00 (0.00) | 0.23 (0.26) | 0.20 (0.24) | 0.26 (0.28) |
| SiB3-exp1 | 0.50 (0.81) | 0.25 (0.66) | 0.04 (0.03) | 1.07 (1.23) | 1.04 (1.21) | 0.16 (0.13) |
| SiB3-exp2 | 0.25 (0.41) | 0.06 (0.17) | 0.32 (0.36) | 0.97 (1.15) | 0.92 (1.13) | 0.26 (0.10) |
| SiB4 | 0.34 (0.91) | 0.12 (0.83) | 0.16 (0.00) | 0.90 (1.04) | 0.86 (1.02) | 0.22 (0.09) |
| ORCHIDEE-exp1 | 0.82 (0.82) | 0.68 (0.67) | 0.00 (0.02) | 1.14 (1.24) | -1.08 (-1.16) | 0.56 (0.67) |
| ORCHIDEE-exp2 | 0.78 (0.79) | 0.61 (0.63) | 0.00 (0.03) | 1.00 (1.20) | -0.95 (-1.12) | 0.51 (0.64) |
| ORCHIDEE-exp3 | 0.59 (0.55) | 0.35 (0.31) | 0.01 (0.20) | 0.63 (0.81) | -0.57 (-0.76) | 0.35 (0.36) |
| BEPS* | X | X | X | X | X | X |
| CLM4.5 | 0.82 (0.85) | 0.68 (0.73) | 0.00 (0.01) | 0.34 (0.35) | -0.31 (-0.31) | 0.21 (0.18) |
| CLM5.0 | (0.96) | (0.92) | (0.00) | (1.09) | (-1.08) | (0.42) |


In summary, TBM-MIP experiments reveal several key factors that can improve or degrade estimates of

spring GPP at Niwot Ridge. For example, adapting model parameters to needleleaf species based on hand-tuning to
tower data and formal data assimilation methods (CLM4.5 and ORCHIDEE-exp3, respectively) improves the
overall magnitude of spring GPP. Likewise, prescribing LAI to a constant value of 4.0 $m^2/m^2$ based on tower
measurements (SiB3-exp2) improves year-to-year variability, while prescribing time variable LAI based on MODIS
data improves spring GPP magnitude (SiB3-exp1). SiB4, which has prognostic rather than prescribed phenology,
represents a compromise in magnitude and variability when looking at the entire record (2001-2018), but is one of
the top performers across all TBM-MIP models over the most recent period (2012-2018).

We did not directly consider changes in canopy structural or biophysical characteristics in our

CARDAMOM experiments. In CARDAMOM, LAI is a prognostic quantity (a function of foliar C and leaf carbon
mass per area). In the absence of LAI observational constraints, CARDAMOM LAI is indirectly informed by the
constraints of time-varying GPP on DALEC2 parameters (see section 2.3). Our results suggest that additional
improvements are possible with careful consideration of in situ measured vegetation parameters.

TBM-MIP experiments also offer insight on important environmental controls and process representation.

Air temperature is an effective constraint of spring GPP onset (CLM4.5, ORCHIDEE-exp1, Figure 7 and Table 2),
but which can be degraded when large scale data assimilation does not account for local- to regional- vegetation
characteristics in parameter optimization (e.g., ORCHIDEE-exp3, Table 2). Water availability appears to be a
secondary but still important driver of spring GPP. While acknowledging the numerous differences between
CLM4.5 and CLM5.0, we find it important to note that plant hydraulic water stress (CLM5.0) shows improved IAV
performance (high correlation, Table 2) over simplified soil moisture stress functions (CLM4.5).  This result further
supports efforts to closely analyze seasonal GPP to locate different environmental controls for future model
improvements.

Our study of the controls of cold temperature on GPP has important implications for modeling seasonal

productivity.  First, future work must evaluate cold temperature limitation at other sites across an array of ecosystem
types.  Additionally, it is important to determine if the temperature thresholds of photosynthesis initiation and
cessation are similar across locations, or unique to ecosystem type and/or site.  Previous studies have had mixed
results, supporting both the use of customized temperature threshold parameters dependent on the site (Tanja et al.,
2003; Chang et al., 2020) or for a general parameter across multiple sites or biome type (Bergeron et al., 2007).
These differences could be due to variations in other variables (e.g., soil temperature, irradiance, etc.) and/or
physiological differences in the vegetation species.  Identifying how photosynthesis temperature thresholds vary
across space and ecosystem type would be beneficial in improving model performance in simulating productivity.
Our model intercomparison study also provide insights on how we may improve our ability to model seasonal GPP.
For example, in Fig. 7b, we see that the ORCHIDEE model growing season starts too early. In the photosynthesis
module of ORCHIDEE, the temperature-dependency of parameters are described by Arrhenius or modified
Arrhenius functions following Medlyn et al. (2002) and Kattge and Knorr (2007). In general, the functions are used
to estimate the potential rates of Rubisco activity and electron transport based on temperature, as these rates are
needed to determine photosynthetic capacity (Medlyn et al., 2002).  The lowest temperature for productivity
mentioned in these studies are 5°C and 11°C, respectively. Additionally, there is a test at the start of the
photosynthesis subroutine that prevents the computation of photosynthesis if the mean temperature over the last 20
days falls below -4°C. For our study, the only ORCHIDEE experiment that uses specific data related to the plant
functional type of this site (OCO-2 SIF data for US-NR1) is ORCHIDEE-exp3.  This experiment improves the
general behavior of the modeled GPP seasonal cycle but does not improve ORCHIDEE's ability to capture the start
of the growing season. So with the future evaluation of cold temperature limitation at other sites and further study of
the potential temperature-influenced bias in the model, then ORCHIDEE (and other process-based models) may
need to improve its photosynthesis temperature-dependency for cold plant functional types. Therefore, we
recommend implementing a cold temperature GPP limitation in a process-based model to confirm its ability to
improve model performance. If we identify (1) how photosynthesis initiation and shutdown varies with temperature
and location, and (2) apply a cold temperature limitation successfully in a process-based model, then we could
expand our analyses to answer bigger Earth science questions. For example, we could use Earth System Model
temperature trends to determine how changing temperature will impact GPP in the future.

While further experiments are needed, these results demonstrate the value of (1) site level data assimilation

for local scale prediction of GPP magnitude and variability, (2) global data assimilation for reducing magnitude
biases, and (3) process formulation for accounting for sensitivity to temperature limitation and water stress. Overall,
these results are encouraging for model-data fusion systems which have developed the capacity to bring together
temporally and spatially resolved functional and structural vegetation components such as LAI, SIF, soil organic
matter, and above- and below-ground biomass (e.g., Bacour et al., 2019; Smith et al., 2020; Bloom et al., 2020).
Joint assimilation of these datasets, coupled with observed meteorological forcing, has potential to introduce more
emergent constraints of vegetation change with respect to environmental change, thus improving overall estimates of
productivity.  Future work will assess the joint impact of SIF, ET, LAI, and biomass data as effective constraints on
light use and water use efficiency (Smith et al., 2020), which is expected to improve the ability of CARDAMOM to
use light with respect to increasing biomass subject to longer growing seasons and heat and water stress.
**4.   Conclusions**

Despite mechanistic advances in ecosystem modeling, it is still a challenge to simulate temporal variations

in GPP.  In an attempt to dissect the environmental controls on GPP in an evergreen needleleaf ecosystem, we
analyzed the impact of temperature on spring (March-May) productivity by implementing a cold temperature GPP
limitation within a model-data fusion system (DALEC2-CARDAMOM).  The cold weather GPP limitation allows
for improved model estimates of mean spring productivity at Niwot Ridge, specifically CARDAMOM's ability to
match the interannual variability observed in tower-derived mean spring GPP.   Furthermore, CARDAMOM is able
to match spring interannual variability between model and tower data outside of the training period.  When
compared to TBM-MIP models, controls that appear to impact model performance include the inclusion of water
stress (e.g., soil moisture) and vegetation parameters (e.g., prescription of LAI). The fact that the cold temperature
limitation does not improve CARDAMOM's annual GPP estimates suggests that other controls (i.e. winter
precipitation) drive GPP variability in other parts of the year, most likely summer (June-September).  The cold
temperature limitation may prove useful in understanding future changes in spring productivity due to changes in
temperature in other ecosystems as well.

**Appendices**

**Appendix A: Model-Data Fusion Methodology**

The DALEC2 model parameter values and state variable initial conditions (henceforth $x$) are optimized using a Bayesian inference approach, where the posterior probability distribution of $x$ given observations O, p(x|O), can be expressed as

$$p(x|o) \propto p(x)L(x|O) \tag{A1}$$

Where p($x$) is the prior probability distribution of $x$, and L($x$|O) is the likelihood of the DALEC parameters and initial conditions given observations **O**. We define the likelihood function as

$$L(x|o) = e^{-\frac{1}{2}\Sigma_i\left(\frac{m_i(x)-o_i}{\sigma}\right)^2} + e^{-\frac{1}{2}\Sigma_a\left(\frac{m_a'(x)-o_a'}{\sigma'}\right)^2}, \tag{A2}$$

where for monthly timestep $i$, $m_i(x)$ and $o_i$ represent monthly modeled GPP (based on parameters **x**) and flux-tower GPP observation, respectively. Following model-data fusion efforts with a spectrum of temporal modes of variability (Desai 2010, Quetin et al., 2020 and Bloom et al., 2020), we extend the cost function to include mean annual model and tower-derived GPP, $m_a(x)$ and $o_a$ respectively) for year = $a$, which allows the GPP cost function to be sensitive to both seasonal and inter-annual components of the flux tower GPP signal. We log-transform modeled and tower-derived GPP values (as done in Bloom & Williams, 2015 and Bloom et al., 2016), which is preferable for characterize model-data residuals between strictly positive quantities (such as GPP). For lack of better uncertainty estimates on monthly and annual flux tower GPP accuracy—including lack of knowledge on GPP error characteristics at monthly timescales, error covariance between individual GPP estimates, model structural error impacts on GPP —we conservatively prescribed uncertainty factor of $\sigma = 2$ for monthly values (roughly ~75%), and σ' = 1.2 (~18%) for annual values; in general we found that these values led to robust agreements between flux tower and DALEC2 GPP variability (model-data mismatch metrics are reported in section 3 of the manuscript).

For all model experiments, we sample the probability of $p(x|o)$, the posterior probability distribution of initial conditions $x$ given observations o, we use four Metropolis-Hastings Markov Chain Monte Carlo (MHMCMC; Haarrio et al. 2001) for $10^8$ iterations; we subsample 1000 parameter vectors $x$, from the latter 50% of each chain (in total 1000 samples x 4 chains = 4000 samples). We test for convergence in the MHMCMC estimates of $x$ using a the Gelman-Rubin convergence diagnostic to measure convergence between the four chains.

**Data Availability**

The Ameriflux US-NR1 data were obtained from: https://ameriflux.lbl.gov/sites/siteinfo/US-NR1 (Blanken et al., 2020). The US-NR1 data used in this study, as well as the CARDAMOM and TBM-MIP outputs are publicly available and provided in .nc file format at http://doi.org/10.5281/zenodo.4928097 .

**Code Availability**

The CARDAMOM code used in this study is available here: https://github.com/CARDAMOM-framework/CARDAMOM_v2.2

**Author Contributions**

SGS, NCP and AAB designed and performed the research. AJN, BR, CB, FM, IB, YZ, BQ, and MS contributed model simulations. DRB, SPB, and PDB contributed observational data. All authors contributed to the writing of the paper and/or revision of the manuscript.

**Supplement**

**Competing Interests**

An author is a member of the editorial board of *Biogeosciences*. The peer-review process was guided by an independent editor, and the authors have also no other competing interests to declare.

**Acknowledgements**

The US-NR1 AmeriFlux site has been supported by the U.S. DOE, Office of Science through the AmeriFlux Management Project (AMP) at Lawrence Berkeley National Laboratory under Award Number 7094866. A portion of this research was carried out at the Jet Propulsion Laboratory, California Institute of Technology, under contract with NASA. Funding from the NASA Earth Science Division Arctic Boreal Vulnerability Experiment (ABoVE) is acknowledged. We acknowledge the MEASUREs program. SGS was partly supported by a University of California, Irvine graduate student fellowship. DRB and BMR were supported by the NASA CMS (80NSSC20K0010) and the NSF Macrosystems Biology and NEON-Enabled Science (1926090) Programs. The National Center for Atmospheric Research (NCAR) is sponsored by NSF. MS was partly supported by the U.S. Department of Energy Office of Science Biological and Environmental Research as part of the Terrestrial Ecosystem Science Program through the Next-Generation Ecosystem Experiments (NGEE) Tropics project. PNNL is operated by Battelle Memorial Institute for the U.S. DOE under contract DE-AC05-76RLO1830.

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
