# Peer review of "Resolving temperature limitation on spring productivity in an evergreen conifer forest using a model-data fusion framework"

_Biogeosciences, 2021_

## Author Comment (AC1)

**Author response to comments on "Resolving temperature limitation on spring productivity in an evergreen conifer forest using a model-data fusion framework"**

We thank the reviewer for their valuable comments. We have made revisions which are detailed in the point-by-point responses below. The comments from the reviewers are in black regular font, our responses are in blue regular font, and text in the revised manuscript are in *blue italic font*.

Exploring the exist but less studied temperature limitation on spring productivity is an important task to improve the realistic of ecosystem models. The authors applied a model-data fusion framework to discuss the uncertainty in such limitation, mainly for the subalpine evergreen forest in Colorado USA. They found that i) the GPP was gradually inhibited at temperature below 6.0 ºC and completely inhibited below -7.1 ºC. ii) cold temperature limitation has an important influence on spring GPP, while not the case for integrated growing season GPP. Other environmental controls, such as precipitation, play a more important role in annual productivity.

Overall, this study seems to be a nice attempt to address this topic, while the most apparent weakness is that this work solely depends on a single site/tree species, whether the conclusion would be fine for subalpine evergreen forest in another region is unclear.

We intended this study to serve as a starting point for analyzing and modeling temperature limitations on GPP in subalpine evergreen ecosystems. We look forward to testing our cold temperature scaling factor in other regions in the future. While not covered in detail in this paper, Famiglietti et al. (2021) compares different versions of the CARDAMOM model, including the cold temperature limitation version used here, to evaluate how model complexity contributes to the predictive skill of the model. This is discussed briefly in Section 3.2. of our manuscript, and we refer the reader to Famiglietti et al. (2021) for more details.

As expected, the analyzed TBM-MIP models have very different performances regarding the reproduce of spring and annual GPP, see Figure 7 and Table. 2. The authors are recommended to explain whether the parameter sets of these models are optimized using a specialized dataset or the observation of this study.

The TBM-MIP experiment parameters were not optimized using the observations of this study in the same way. Most of the experiments were not optimized and use default parameters, some of the models were optimized using prescribed site-based (US-NR1) characteristics (e.g., CLM), and others were optimized using OCO-2 SIF data (e.g., ORCHIDEE). We have added the following text to Section 2.5. to further clarify how the model parameters are optimized:

*Most of the TBM-MIP model experiments were run with default parameters (BEPS, CLM50, SiB3, SiB4, ORCHIDEE-exp1 and exp2). The other experiments were optimized in the following ways: either a) parameters were hand-tuned based on the US-NR1 data (CLM45) or b) the parameters were optimized using OCO-2 SIF data (ORCHIDEE-exp3). For more details on the parameterization of the TBM-SIF experiments, we refer the reader to Parazoo et al. (2020). The use of these models provides insight into the spread in model structures and the use of their default parameters.*

---

## Author Comment (AC2)

**Author response to comments on "Resolving temperature limitation on spring productivity in an evergreen conifer forest using a model-data fusion framework"**

General comment:
The introduction of a cold temperature scaling function into CARDAMOM significantly improves CARDMOM's ability to represent the interannual variability observed in the tower-derived mean spring GPP though it results in a slight degradation of CARADMOM in representing the seasonal cycle of GPP at an evergreen needleleaf site. The cold temperature scaling function doesn't improve CARDAMOM's ability in estimating summer or annual GPP at the site. The result suggests other environment controls might impact summer and annual GPP variability while CARDAMOM doesn't include that scheme. It's a good paper in proving that the added cold temperature scaling function does make CARDAMOM in capturing the interannual variability of observed mean spring GPP which could not be seen in CARDAMOM before the cold temperature scaling function is added.

We thank the reviewer for their valuable comments. We have made revisions which are detailed in the point-by-point responses below. The comments from the reviewers are in black regular font, our responses are in blue regular font, and text to be included in the revised manuscript are in *blue italic font*.

Specific comments:
C1#: lines 160-162. Fig. 2 only shows the scatter points of spring GPP and spring air temperature. It doesn't show the scatter points of spring GPP and summer air temperature or scatter points of spring GPP and winter precipitation. I suppose you actually intend to might use a figure showing the relationship between the environmental forcings and mean spring temperature instead of mean annual GPP as shown in Figure S2.

We did not initially intend to use a figure showing the relationship between the environmental forcings and mean spring GPP. In the interest of brevity, we felt it was sufficient to state the correlations with the different environmental forcings for spring GPP, as temperature is the focus of our analysis.  To address this comment, we have added an additional supplementary figure, like Figure 2, showing the relationship between mean spring GPP and the other environmental forcings (i.e., summer air temperature, winter precipitation).  *

C2#: lines 164-167. I don't understand the reason why you investigate the correlation between the environmental forcings and mean annual GPP here. I think if you move the result described here to somewhere near line 377, it will make the story well connected. Otherwise, the reader has to go back and forth to understand your point. Some might even forget what you already described here when they read the lines in the far behind and get confused there.

We think there is value in discussing the correlation between environmental forcings and annual GPP in this section. This analysis contributed to the development of cold temperature scaling factor and supports the storyline of this paper.  To address this comment, we have summarized the results discussed in lines 164-167 again near line 377. We have added the following text:

*Since annual average GPP appears to be more dependent on winter precipitation/snowpack (Pearson's linear r =0.63, Fig. S2A), future work will include improving model predictability of late season productivity and quantifying temperature-water effects on carbon uptake.*

C3#: lines 205-207. What is the $T_{min}(t)$?

$T_{min}(t)$ is the observed minimum temperature at Niwot Ridge at time *t*. We have added this definition to the manuscript at line 209.

C4#: The equation (2) will scale GPP between photosynthesis shutdown (0) and photosynthesis initiation (nominal GPP) when $T_{min}(t)$ is between $T_0$ and $T_g$. Does the physical scheme as described by the equation exist in reality? Or Whether in reality does such GPP between photosynthesis shutdown and initiation exist? Or the equation is just an empirical equation, and it doesn't represent the actual process at all.

We believe equation (2) represents the impact of temperature on plant hydraulics and photosynthetic activity. We have added this discussion to the manuscript (Section 2.4):

*Equation (2) may physically represent changes in plant hydraulics and photosynthetic activity due to changing temperature in the spring. As temperature increases, evergreen stems slowly thaw out, which enables the trees to access available soil moisture and slowly reactivate their carbon and water exchange processes (Mayr et al. 2014; Bowling et al. 2018). Temperature also impacts the reactivation of photosynthetic activity after winter dormancy (Oquist & Huner 2003, Tanja et al. 2003). For example, fluctuating temperature in the spring has been shown to limit and sometimes reverse the activation of biochemical processes needed for photosynthesis recovery (Ensminger et al. 2004). Exposure to cold temperature, when combined with increased irradiance in the spring, can also damage evergreen trees (Oquist & Huner 2003, Yang et al. 2020), therefore disrupting $CO_2$ assimilation. These processes may be captured in this cold-temperature scaling factor added to CARDAMOM.*

C5#: line 225. 'parameter optimization, and data assimilation'. To my understanding, the Model-Data Fusion proposed by Bloom and Williams (2014) which is used in your study is a framework to optimize the parameters in DALEC model and it is a parameter optimization method instead of data assimilation. Data assimilation and parameter optimization are two different methods to reduce model uncertainty. You're not using the 'data assimilation' method in your study. The so-called data assimilation mentioned here is actually parameter optimization. You could read some papers about data assimilation to know the difference between the two methods.

We have rephrased line 225 to the following:

*These four experiments serve to evaluate the sensitivity of modelled GPP at Niwot Ridge to cold temperature limitation and parameter optimization.*

C6#: lines 249-253. A table that shows the difference between the within-model experiments described here is better than the text for readers to remember and check the difference when they read the results and discussion later.

In the interest of brevity and to reduce the number of large tables, we chose to not include a table comparing the within model-experiments. We refer the readers to the Parazoo et al. (2020) study, Table 1 to see full descriptions of the TBM-MIP model experiments, and briefly summarize the key differences in the models.

C7#: Do you have any insight that whether the cold temperature scaling function will also improve the fall GPP simulation? I bet so.

Yes, the cold temperature scaling function could potentially improve the simulation of fall GPP. We have added the following text to the discussion (Section 3.2):

*Temperature is important in both the reactivation of photosynthetic activity in the spring and the wind down of activity in the fall (Flynn & Wolkovich 2018, Stinziano et al. 2017). Therefore, we anticipate that the cold temperature scaling function may also improve our ability to model fall productivity. However, other factors such as water availability and photoperiod must also be considered (Bauerle et al. 2012; Stinziano et al. 2015). Future studies at Niwot Ridge and other sites should investigate the role of these factors (temperature, water, photoperiod) in regulating fall GPP and how we can represent these processes in CARDAMOM.*

C8#: lines 388-390. The model intercomparison provides the direction or strategy which you can take to further improve the annual GPP simulation at the site. The statement is easy for readers to understand why you do the model intercomparison here. Instead, the original sentence is hard to understand especially 'discern commom environmental controls in …' because the difference of environmental controls on different models is so subtle at the point until you reveal them later in lines 450-457.

For clarity, we have rephrased the sentence on lines 388-390 to the following:

*As such, we emphasize that our model comparison is not a strict assessment of performance, but rather an attempt to learn how model simulation of GPP at an evergreen needleleaf site can be improved.*

C9#: lines 455-456. 'high correlation and reduced error'. I can see the higher correlation between CLM5.0 and the observations from both Fig7(a) and Table2, but the RMSE and MBE are higher for CLM5.0 compared with CLM4.5. The reduced error means RMSE or MBE? or something else? If the reduced error here means RMSE or MBE, it's not consistent with what is shown in both Fig7(a) and Table2.

We incorrectly stated that the error (RMSE/MBE) in mean spring GPP was reduced for CLM5.0 compared to CLM4.5. We have corrected this in the manuscript to the following:

*While acknowledging the numerous differences between CLM4.5 and CLM5.0, we find it important to note that plant hydraulic water stress (CLM5.0) shows improved IAV performance (high correlation) over simplified soil moisture stress functions (CLM4.5).*

C10#: line 474. 'improved model estimates of productivity at Niwot Ridge' in what sense? if compared with CARD, CARDcold is actually sligthly worse in representing the accuracy of seasonal cycle of GPP. You'd better to address in what sense the improvement is. Otherwise, it's not accurate.

We have rephrased the sentence on line 474 to emphasize CARDcold's improvements in modeling mean spring GPP at Niwot Ridge, especially the model's ability to match spring interannual variability (IAV).

*The cold weather GPP limitation allows for improved model estimates of mean spring productivity at Niwot Ridge, specifically CARDAMOM's ability to match the interannual variability observed in tower-derived spring GPP.*

C11#: line 482. Instead of 'Western US', subalpine temperate forests might be more reasonable.

We have changed the reference to 'Western US' on line 482 to 'subalpine temperate forests' as recommended by the reviewer.

Technical corrections:
T1#: line 194. Figure 3. The x-labels of the three sub-figures should be maximum, minimum, and mean monthly air temperature respectively as described in lines 181-183 instead of spring air temperature.

We have corrected this error in the Figure 3 x-labels.

T2#: line 414. 'b.) monthly GPP'. 'monthly GPP from 2015 to 2018' is clear. As I read it, I have to go to lines 256-257 to make sure all the data are from the same period.

We have changed line 414 to say "*monthly GPP from 2015-2018'* as recommended by the reviewer to make it more clear which data we are plotting in the figure.

T3#: lines 440-442. The description about SIB3-exp1 and SIB3-exp2 here is contrary to that in line 250 and lines 407-408.

SiB3-exp1 is prescribed with monthly values, and SiB3-exp2 LAI is fixed at 4.0 $m^2/m^2$. This is correctly states in line 250. We have corrected lines 407-408 and 440-442 to align with this statement.

T4#: lines 450-453. Could you please mention the figure or any table at the end of the sentence in lines 450-453 to support your point? It will be easier for us to follow your point if you add the figure or table from which you conclude your point.

We have added references to Figure 7 and Table 2 to support the point made in lines 450-457.

---

## Author Response (AR2)

Dear Editor,

Please find attached to this document our point-by-point responses to reviewer comments for manuscript BG-2021-152, "Resolving temperature limitation on spring productivity in an evergreen conifer forest using a model-data fusion framework" by Stettz et al. In response to the reviewer comments, we have made several revisions to the manuscript. These revisions include a description of the potential relationship between our cold temperature scaling factor and biophysical mechanisms, as well as the implications of the cold temperature limitation for modeling fall productivity. We have also added one supplementary figure showing the relationship between other meteorological forcing (i.e., mean summer air temperature and mean winter precipitation) and mean spring GPP as suggested by Reviewer #2. As requested by the editor, we have added an additional paragraph to Section 3.3, discussing the implications of our results for other modeling groups and locations to broaden the usefulness of our results. Additionally, we have added text on the representation of soil moisture stress in CARDAMOM.

We also note that while revising the manuscript, we noticed a small issue concerning the magnitude of average temperature observations plotted in Figures 1-3. We have fixed this issue, which resulted in only minor changes to the figures. As such, all the key results remain the same. We believe that the changes made have substantially improved the manuscript and broadened its impact to a wider array of readers. We have included the revised manuscript, as well as a marked-up version of the manuscript showing tracked changes, with this resubmission. Thank you for your consideration.

Sincerely,
Stephanie Stettz (corresponding author)
sstettz@uci.edu